# Optimising pacemaker therapy and medical therapy in pacemaker patients for heart failure: protocol for the OPT-PACE randomised controlled trial

Maria F Paton,[1] John Gierula,[1] Haqeel A Jamil,[1] Judith E Lowry,[1] Rowena Byrom,[1] Richard G Gillott,[2] Hemant Chumun,[1] Richard M Cubbon,[1] David A Cairns,[3] Deborah D Stocken,[3] Mark T Kearney,[1] Klaus K Witte[1]

¹Leeds Institute of Cardiovascular and Metabolic Medicine, University of Leeds, Leeds, UK
²Department of Cardiology, Leeds Teaching Hospitals NHS Trust, Leeds, UK
³Leeds Institute of Clinical Trials Research, University of Leeds, Leeds, UK

**Correspondence to**
Dr Klaus K Witte;
k.k.witte@leeds.ac.uk

## ABSTRACT

**Introduction** Permanent artificial pacemaker implantation is a safe and effective treatment for bradycardia and is associated with extended longevity and improved quality of life. However, the most common long-term complication of standard pacemaker therapy is pacemaker-associated heart failure. Pacemaker follow-up is potentially an opportunity to screen for heart failure to assess and optimise patient devices and medical therapy.

**Methods and analysis** The study is a multicentre, phase-3 randomised trial. The 1200 participants will be people who have a permanent pacemaker for bradycardia for at least 12 months, randomly assigned to undergo a transthoracic echocardiogram with their pacemaker check, thereby tailoring their management directed by left ventricular function or the pacemaker check alone, continuing with routine follow-up. The primary outcome measure is time to all-cause mortality or heart failure hospitalisation. Secondary outcomes include external validation of our risk stratification model to predict onset of heart failure and quality of life assessment.

**Ethics and Dissemination** The trial design and protocol have received national ethical approval (12/YH/0487). The results of this randomised trial will be published in international peer-reviewed journals, communicated to healthcare professionals and patient involvement groups and highlighted using social media campaigns.

**Trial registration number** NCT01819662.

## INTRODUCTION

Permanent artificial pacemaker implantation is a safe and effective treatment for bradycardia[1] and is associated with extended longevity[2] and improved quality of life.[3] An estimated 350 000 people in the UK have a pacemaker, with over 40 000 new implants per year.

However, long-term right ventricular (RV) pacing has been linked to adverse left ventricular (LV) remodelling,[4 5] such that the most common long-term complication of standard pacemaker therapy is pacemaker-associated

### Strengths and limitations of this study

► OPTimising PACEmaker therapy is a trial independently funded by the National Institute for Health Research.
► Stratification of the participant randomisation is not adopted in this trial due to the large sample size.
► Participant and practitioner blinding are not plausible.
► The control arm consists of current standard of care ensuring trial results are reflective and generalizable.

chronic heart failure (CHF) due to left ventricular systolic dysfunction (LVSD).[6–8] While up to 2%–3% of the general population have CHF, the condition is much more common in pacemaker patients with a prevalence up to 50%[7 9] and 12% of people admitted with acute decompensated heart failure (HF) (4% with a de novo admission for HF) have a pacemaker.[10]

It is clear that RV pacing alone can lead to LVSD; however, the risk is especially high in people requiring a high proportion of ventricular pacing, those with diabetes mellitus, previous myocardial infarction and raised creatinine.[11] Data examining the relationship between pacemaker use and cardiac dysfunction predominantly originate from retrospective cross-sectional analysis or secondary analyses.[12–15] We provided the first evidence that reducing RV pacing through careful reprogramming is associated with an improvement in LV function without affecting quality of life or functional capacity,[9] confirming somewhat the suggestion that RV pacing is contributory to LV dysfunction in pacemaker patients and not just a bystander in a multimorbid patient population.

Although there is increasing recognition of the axis of RV pacing and LVSD, it

**BMJ**

remains the case that HF is frequently overlooked in the pacemaker population with symptoms often ascribed to pacemaker syndrome, chronotropic incompetence or comorbidities.[6] HF and undiagnosed LV dysfunction have a major effect on mortality and morbidity among pacemaker patients.[16–18] Since HF management accounts for approximately 2% of the healthcare budget in many developed countries, more than 60% of which relates to hospitalisation costs,[19] optimal management of people with HF whether or not they have a pacemaker is a key strategy for controlling healthcare costs while increasing quality of life.[20]

There are, however, no data on the potential benefits of screening people with pacemakers for HF, and while there are now a series of algorithms to limit RV pacing,[21–23] no clear strategy for the use of these is outlined in current guidelines. This situation is compounded by a lack of published evidence around the benefits of medical management of pacemaker-related HF since people with pacemakers were excluded from most of the large HF studies.

Prospective randomised studies of pacemaker follow-up programmes to risk stratify patients and optimise management have not yet been of adequate size and duration to assess the clinical effectiveness of such an approach in a real-world general healthcare setting. Pacemaker follow-up is a rarely realised opportunity to screen for HF and assess and optimise patient devices and medical therapy.[24]

The OPTimising PACEmaker therapy (OPT-PACE) trial has been designed to address these issues and quantify time to all-cause mortality or HF hospitalisation, as well as validate a HF risk stratification model based on simple variables available in pacemaker clinic. Secondary aims are to establish whether N-terminal prohormone of brain natriuretic peptide (NT-proBNP) improves this risk stratification model. Subsequently, we aim to provide data on the effects of establishing the presence of LVSD on medical therapy, device programming and patient quality of life.

## Pilot data
### Risk model
The design of the OPT-PACE trial is informed by an observational cohort[11] including 491 patients listed for a pacemaker generator replacement in a single tertiary centre (Leeds Teaching Hospitals NHS Trust) invited for pacing therapy information, diagnostic pacing data and an echocardiogram.

Of this cohort, 40% had a left ventricular ejection fraction (LVEF) of <50%, which was much higher (59%) in those with >80% RV pacing (p<0.001), demonstrating that patients with RV pacemakers have a high prevalence of LVSD. After a mean follow-up time of 668 days, 56 patients (12%) had died or been hospitalised for HF.

Multivariable analyses identified a number of simple clinical variables; high percentage RV pacing, high serum creatinine and previous myocardial infarction as potential independent predictors of LVSD to identify patients who may benefit from a more comprehensive review.[11] This risk model requires external validation planned within the OPT-PACE trial.

## Intervention development
The design of the pacemaker optimisation intervention used in the OPT-PACE trial is based on data from an observational cohort of 66 patients, referred for pacemaker generator replacement, recruited consecutively from a single tertiary centre (Leeds Teaching Hospitals NHS Trust).[9] All patients recruited had been referred for pacemaker generator replacement. Exclusion criteria were inability to consent, underlying complete heart block, a life expectancy of <1 year in the opinion of the clinician (eg, terminal malignancy), the presence of a device-related complication or those with known LVEF <50%.

In patients recruited with avoidable RV pacing, a prespecified protocol was followed to reduce this as outlined in figure 1, including a reduction in the diurnal base rate (BR) to 50 beats/min with nocturnal rate or hysteresis to 40 beats/min along with the deactivation of rate-adaptive pacing. In addition, where sinus rhythm and heart block had been documented, the atrioventricular delays were extended or a device with RV avoidance algorithms was implanted. The patients were reassessed 6 months post randomisation. The primary endpoint was LVEF at follow-up calculated as an average over three non-paced, normally conducted beats using modified Simpson's method as per American Society of Echocardiogrpahy guidelines.[25] The protocol was tolerated in all but two subjects in whom rate-responsive pacing was reactivated. Nevertheless, application of the protocol reduced absolute RV pacing percentage by a mean of 49% (95% CI 41% to 57%; p<0.0001) from baseline and resulted in a mean absolute improvement in LVEF of 6% (95% CI 2% to 8%; p<0.0001). This occurred without reduction in exercise capacity, NT-proBNP or quality of life. An association was seen between the reduction in RV pacing and magnitude of change in LVEF (p=0.04). The intervention was considered safe to use without adaptation in OPT-PACE.

## METHODS AND ANALYSIS
OPT-PACE is a multicentre, randomised, non-blinded parallel trial funded by the National Institute for Health Research (NIHR) in the UK (NIHR-CS-2012-032) and registered with the Clinical Trials registry. The trial design and protocol have received approval from necessary regulatory and ethics boards as well as an independent trial steering committee.

The trial will recruit 1200 patients, which began in June 2013 and all patients will be followed up for a minimum of 12 months.

Trial participation may be terminated via voluntary refusal to continue at any time or through significant clinical deterioration assessed by medical staff. Participants

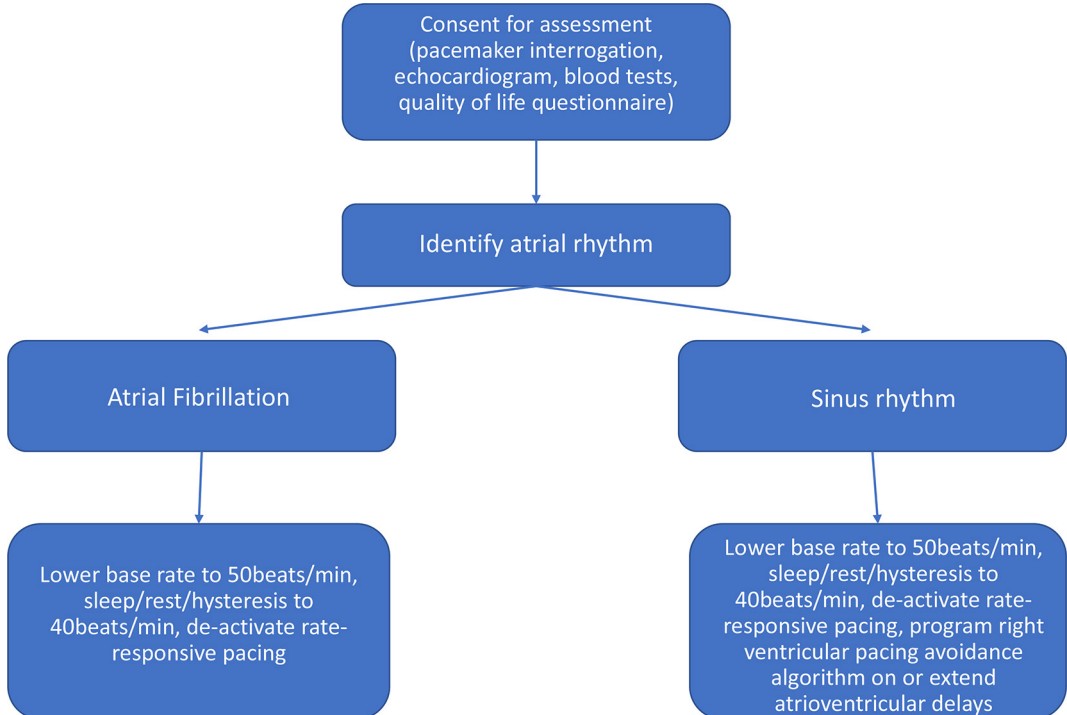

**Figure 1** Ventricular pacing avoidance protocol.[9]

suffering from trial-related problems or adverse events (AEs) may obtain medical treatment as compensation, and the trial is insured to compensate.

## Patients
Patients are recruited from one tertiary centre (Leeds Teaching Hospitals Trust) and two district centres (Harrogate District Foundation Trust and Bradford District Trust). Participants will have an implantable pacemaker for bradycardia for at least 12 months according to any indication in current accepted clinical guidelines[26] and be able to provide written informed consent. Both atrial and ventricular pacing burdens will be documented. Participants will be excluded if they have an implantable cardioverter defibrillator or cardiac resynchronisation device, are less than 18 years old, pregnant, awaiting heart transplantation or have a severe comorbidity with life expectancy of <1 year. We will also exclude people with significant cognitive impairment and any already under the care of HF services.

## Randomisation and interventions
Potential participants will receive a thorough explanation of the process of the trial, be notified of appointments via post and associated travel costs will be available to maximise compliance. Potential participants will be asked to voluntarily provide written informed consent prior to randomisation in line with the Declaration of Helsinki 2002.

Consecutive pacemaker clinic attendees agreeing to participate will be enrolled and randomised centrally using a study-specific electronic system. Patients are randomised in a 1:1 ratio to (1) standard care pathway or (2) interventional pathway (figure 2). Usual care pathways follow National Institute for Health and Care Excellence (NICE) guidelines,[27] and patients will undergo routine, usually annual, pacemaker follow-up. Interventional pathways consist of enhanced or optimised care predetermined by each recruiting site. The echocardiographically informed pathways will consist of either a primary care driven management plan based on the echocardiogram that will be forwarded to the primary care team (enhanced care, n=300) if recruited in one of the district centres or a comprehensive care package consisting of optimised programming and medical therapy co-ordinated by a regional HF service (optimised care, n=300) if recruited in the tertiary centre. This interventional package will include the RV pacing avoidance algorithm (figure 1) and in those with LVSD (LVEF <50%) medical therapy following the NICE guidelines for the management of CHF.[28] In the presence of LVSD, patients recruited at the tertiary centre will be referred to an outpatient HF clinic (optimised care), and in the district centres an echocardiogram report will be sent to the patients' general practitioner (enhanced care).

## Blinding
Practitioners and participants in this trial cannot be blinded to allocation of treatment due to the nature of the intervention. By not participating in baseline assessments or medical and device optimisation and having no prior knowledge of participant allocation, the outcome assessor will remain blinded and thereby safe from detection bias.[29] Unblinding of the assessors will only be permitted in certain circumstances, for example, when

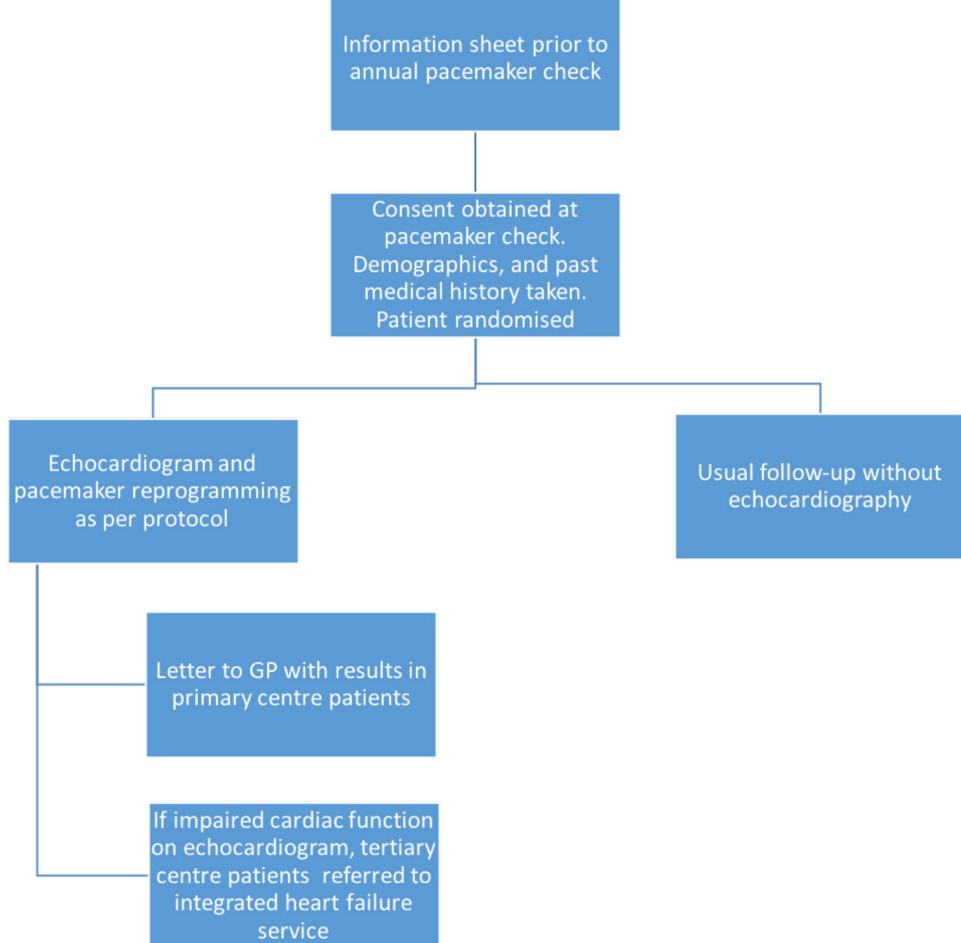

**Figure 2** OPT-PACE study protocol.

the information is demanded to ensure appropriate management of the participants (eg, serious AEs).

## Outcome measures

The primary outcome measure is a combined endpoint of time from randomisation to date of first event of all-cause mortality or HF hospitalisation as assessed by the Endpoint Review Committee. The Endpoint Review Committee will ensure the quality of conduct and assess protocol deviations due to the low risk associated with intervention (echocardiography). They will monitor the safety of the trial and have unblinded access to efficacy data should it be required.

The main secondary outcome measure is the validation of a model to risk stratify people with a pacemaker for prevalent or future HF using simple clinical and pacing variables to predict the presence of LVSD at baseline and subsequently the combined outcome of HF hospitalisation and mortality at 1 year. Validation will be achieved by assessing against prespecified criteria, formed from the CIs of the initial pilot data analysis of the model.[9]

Other secondary outcome measures are to determine (1) the utility of BNP testing as a way to improve risk score for prevalent LVSD and future admissions, (2) hospitalisation rates and mortality across randomised arms, (3) utility of a diagnosis of cardiac dysfunction on achieving standard of care medications and doses and (4) quality of life as measured by the EuroQol 5-Dimension (EQ-5D) at baseline and 12-month postenrollment.

All AEs related to the interventions will be recorded and monitored until resolved. Medical personnel will decide if continued trial participation is feasible based on these reports, with the final decision made by the participant.

## Data collection

A clinical research co-ordinator (CRC) will collect information on demographic characteristics such as gender, age, vital signs, height and weight and medication history. The patients' medical history will be taken along with pacemaker data (implant date, implanted device, programmed pacemaker mode, BR and cumulative pacing percentages), and blood draw will be obtained. Biological specimens will be analysed and stored in the onsite research pathology laboratory. Patients randomised to intervention will undergo a baseline transthoracic echocardiogram, which will be assessed by the CRC. All data will be stored anonymously and will be locked once complete.

To ensure protocol adherence, a single research team will undertake all study-specific activity and be regularly monitored.

Digital follow-up will occur at 12-month postrecruitment to record mortality and hospitalisation data and changes to medical therapy. Patient data regarding cardiac transplatation or device upgrade will be recorded and subsequently reported.

## Sample size

The trial is designed to detect a reduction in hospitalisation or mortality rate of 7.5% at 1 year in patients identified with cardiac dysfunction from 15% anticipated in patients randomised to the standard pathway.[9] Given approximately one third of patients in both randomised arms are estimated to have cardiac dysfunction, a 7.5% reduction would be diluted to a 9% overall reduction in the enhanced pathway arm. To detect a reduction in events from 15% to 9% (equivalent to a HR equal to 0.58) using log-rank analysis with an overall type 1 error rate of 0.05 (two-sided analysis) and a power of 0.90 requires a total of 146 events to be observed in at least 1070 participants (nQuery Advisor assuming 18-month recruitment) inflated to 1200 in anticipation of minimal drop-out.

## Statistical analysis plan

Primary analysis will be performed using the prespecified endpoints in the intention-to-treat population (all randomised patients). Time to event will be summarised across randomised groups using Kaplan-Meier estimates, HRs and comparison across groups using log-rank tests, reporting 12-month survival.

Secondary analyses will estimate adjusted treatment effects through Cox proportional hazards regression model to describe the influence of baseline patient characteristics on outcome. Factors to explore include age, gender, site, comorbidities, type of device, New York Heart Association (NYHA) class, medication, blood measures and underlying rhythm. A post hoc exploratory cost-effectiveness analysis will be conducted.

Quality of life data is collected at two time points and will be summarised graphically, conditional on patient survival. Quality-adjusted time to event analysis may be appropriate to report quality-adjusted estimates of 12-month survival.

External validation of the risk model will be based on fitting the published model to the OPT-PACE trial data to report its goodness of fit. Data will be pooled, and stratified by study, to remodel parameter estimates with the aim of improving precision.

## Patient and public involvement

The research question developed as a large number of pacemaker patients was seen in HF clinic with deterioration in their ventricular function and symptoms who were followed-up annually in pacemaker clinic and yet, these had not been detected. The study was initially discussed with a well-established local patient and public

involvement (PPI) advisory group (AG) consisting of cardiovascular patients and their families. The AG particularly felt that undergoing all study assessments in 1 day was appealing to potential participants but that the intervention was not a large burden and that it was more important to know the effect on patient survival and the number of times patients were hospitalised, not just whether there was an improvement in heart function from optimised care in terms of outcome measures. Once the final protocol was established, it was reviewed by the PPI-AG along with patient information sheets to ensure the plain English was at an appropriate level. Regular updates will be presented regarding recruitment and any AEs. Study results will be discussed and an appropriate method of dissemination will be designed together between the AG and the researchers to deliver these to participants.

## Ethics and dissemination

Amendments will be documented. Trial results will be communicated on a local level to healthcare professionals and patient involvement groups. Results will be published in international peer-reviewed journals with authorship as determined by the international committee of medical journal editors' guidelines[30] and highlighted using social media campaigns. The National Institute of Health Research will be made aware prior to publication but no publication restrictions apply.

## Discussion

Patients with permanent pacemakers are at increased risk of prevalent HF with a reduced quality of life and incident HF leading to hospitalisations and a poorer prognosis. Despite increasing recognition of this, and the development of device-based options to limit unnecessary ventricular pacing, patients with permanent pacemakers rarely undergo any assessment to identify prevalent HF or their future risk of HF. Pacemaker follow-up services have remained resolutely technical with intervals based on historical risk of pacemaker device failure rather than focusing on patient requirements in the face of negligible pacemaker failure rates and extended battery longevity.

A validated tool combining clinical and technical data by which patients' risk of the one major complication of pacemaker therapy—HF—could be assessed, allowing individualised follow-up intervals has the potential to substantially improve patient experience while also offering the opportunity to focus screening and lead to the initiation of optimal therapy. Although there are several clinical and pacemaker-related features linked to the presence of cardiac dysfunction, no validated risk score exists to select those that should be referred for assessment.

The management pathways used in this trial will offer individualised healthcare and establish the relative benefits of primary, secondary and tertiary expertise and interventions. Our previous data have shown that reprogramming to avoid RV pacing is safe and well tolerated by patients. Combining this with medical therapy for HF

in collaboration with a HF service as a new model of care for people with pacemakers is feasible and could improve their quality of life and overall prognosis while also being highly cost-effective. We aim to test the clinical effectiveness of this approach when applied to a large number of patients at multiple sites.

## CONCLUSIONS

OPT-PACE is an important randomised controlled trial, supported by non-industry funding, that will investigate whether assessing LV function to inform pacemaker programming and medical decisions can improve the morbidity and mortality in pacemaker patients compared with usual care.

The results will have implications for the stratification of pacemaker patients and the organisation of diagnostic and therapeutic pathways to reduce the incidence of HF in this population.

**Acknowledgements** The authors would like to express their appreciation for the valuable contributions of participants with a pacemaker who are, or will, participate in this trial, the PPI-AG for their practical and insightful suggestions and for the consistent administrative support of Mrs Andrea Marchant and Miss Lisa Trueman.

**Contributors** KKW, MFP, JG, MTK and RMC researched the topic and devised the study. MFP and KKW provided the first draft of the manuscript. DDS and DAC provided statistical oversight and RGG managed data storage and aided analysis. MFP, JG, HAJ, JEL, RB, HC, RMC, MTK and KKW contributed equally to data collection and all authors contributed to manuscript preparation.

**Funding** The research is supported by the National Institute for Health Research (NIHR) infrastructure at Leeds, and an NIHR clinician scientist fellowship (NIHR-CS-2012-032). The views expressed are those of the author(s) and not necessarily those of the NHS, the NIHR or the Department of Health. This research was supported by the National Institute for Health Research Leeds Clinical Research Facility.

**Competing interests** MFP holds an NIHR clinical doctoral fellowship outside of this work. JG holds an NIHR postdoctoral fellowship outside of this work. RMC holds a fellowship with the BHF. MTK holds a BHF chair. KW holds an NIHR clinician scientist award.

**Patient consent for publication** Not required.

**Ethics approval** The trial design and protocol have received ethical approval from the South Yorkshire research ethics committee (12/YH/0487).

**Provenance and peer review** Not commissioned; externally peer reviewed.

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
