## [Reviewer comments · BMJ Open]

ARTICLE DETAILS

TITLE (PROVISIONAL)	Optimising pacemaker therapy and medical therapy in pacemaker patients for heart failure – protocol for the OPT-pace randomised controlled trial
AUTHORS	Paton, Maria; Gierula, John; Jamil, Haqeel; Lowry, Judith; Byrom, Rowena; Gillott, Richard; Chumun, Hemant; Cubbon, R.M.; Cairns, David; Stocken, Deborah; Kearney, Mark; Witte, Klaus

VERSION 1 – REVIEW

REVIEWER	Haran Burri University Hospital of Geneva, Switzerland
REVIEW RETURNED	02-Jan-2019

GENERAL COMMENTS	Paton et al. present the R&D manuscript of the OPT-PACE study, which is of clinical interest. I have the following comments : 1. Will patient inclusion take into account the percentage of ventricular pacing? I would not expect any impact of optimization in patients implanted for sinus dysfunction with a negligible percentage of ventricular pacing. If %VP is not taken into account, the authors should comment as to why not.2. The delay of one year between device implantation and screening for inclusion is relatively short. Why did the authors not simply evaluate all patients implanted since e.g. > 2 years? The likelihood of finding LV dysfunction (ie. Which is actionable) is likely to be higher, and this strategy would speed up inclusion.3. Line 45: it seems that the enhanced care pathway in fact consists of two different regimens. This may impact results (I would expect patients to benefit more from “optimized care” than “enhanced care”). Also, it is somewhat confusing that the enhanced care pathway includes these two denominations (I suggest that “enhanced care” referring to the primary care team is labelled differently).
---

REVIEWER	Marianne Gillam University of South Australia, Australia
REVIEW RETURNED	10-Feb-2019

GENERAL COMMENTS	Thank you for the opportunity to review this protocol which describes an RCT of pacemaker patients allocated to either enhanced care based on transthoracic echocardiogram or normal care. The trial has been registered at ClinicalTrials.gov, obtained ethics approval and is funded by the National Institute for Health Research.
---

	I have some concerns with the protocol, including the following:  - According to the information on ClinicalTrials.gov the trial started August 2013, the actual primary completion date was October 2017, and the latest recorded anticipated study completion date is this month (February 2019). The anticipated study completion date has been moved forward several times after the primary completion date. Publication of the protocol at this late stage when the trial is basically finished does not ensure increased transparency regarding deviation from the protocol which is a major reason for publishing a protocol. - In the protocol the enrolment number is stated to be 1200 participants. According to information on ClinicalTrials.gov, 1793 patients were enrolled before completion (October 2017), but in earlier study record versions 2100 participants were anticipated. Is this discrepancy an error or was the sample size adjusted during the trial period? - There is lack of detail in the protocol e.g. It is not clear how it was decided which patients received enhanced care or optimised care in the intervention group. What was end of follow up - how will data from patients who had heart transplants or changed to other CIEDs be analysed? Is the primary outcome a combined end-point (time to death and heart failure hospitalisation) or either heart failure hospitalisation or death? If heart failure hospitalisations will be analysed separately, I would suggest using competing risk methods, see Austin, P. C., D. S. Lee and J. P. Fine (2016). "Introduction to the Analysis of Survival Data in the Presence of Competing Risks." Circulation 133(6): 601-609.
--	---

VERSION 1 – AUTHOR RESPONSE

Referee 1:

We are grateful for the supportive comments and suggestions which have undoubtedly improved the manuscript. We have made the following changes in response.

Will patient inclusion take into account the percentage of ventricular pacing? I would not expect any impact of optimization in patients implanted for sinus dysfunction with a negligible percentage of ventricular pacing. If %VP is not taken into account, the authors should comment as to why not.

Thank you for this query. Ventricular pacing burden will be recorded but is not a stratification factor for patient inclusion as pilot data we have collected shows a considerable percentage of patients with 100% ventricular pacing at implant are subsequently minimally paced in the ventricle, even after 6 weeks. Additionally, although it has been shown increased ventricular pacing can increase the risk of left ventricular systolic dysfunction, not all patients with a high pacing percentage develop impairment. As a consequence our aim is to undertake a real-world trial of all patients implanted with bradycardia devices. We have now clarified this in lines 198-201 and 267.

“Participants will have an implantable pacemaker for bradycardia for at least 12 months according to any indication in current accepted clinical guidelines [27] and be able to provide written informed consent. Both atrial and ventricular pacing burdens will be documented.”

“The patients’ medical history will be taken along with pacemaker data (implant date, implanted device, programmed pacemaker mode, base rate, and cumulative pacing percentages), and blood draw will be obtained.”

The delay of one year between device implantation and screening for inclusion is relatively short. Why did the authors not simply evaluate all patients implanted since e.g. > 2 years? The likelihood of finding LV dysfunction (ie. Which is actionable) is likely to be higher, and this strategy would speed up inclusion.

Thank you for bringing this to our attention. Any patients implanted with a bradycardia device with at least one year post implant will fulfil the inclusion criteria. From one year post implant, patients are unlikely to develop other pacing-associated complications so this increases the likelihood patients recruited are receiving stable pacemaker therapy prior to participation as well as demonstrating a real-world pacing population.

“Participants will have an implantable pacemaker for bradycardia for at least 12 months according to any indication in current accepted clinical guidelines [27]”

Line 45: it seems that the enhanced care pathway in fact consists of two different regimens. This may impact results (I would expect patients to benefit more from “optimized care” than “enhanced care”). Also, it is somewhat confusing that the enhanced care pathway includes these two denominations (I suggest that “enhanced care” referring to the primary care team is labelled differently).

Thank you for this comment and the opportunity to clarify. We have now consistently termed the interventional pathways as such based upon your advice. Optimised care refers to management within heart failure clinics, enhanced care refers to management through the primary care team, which we believe describes the pathway fairly. We have clarified these terms further in the protocol text, lines 214-228 as seen below.

“Patients are randomised in a 1:1 ratio to 1) standard care pathway or 2) interventional pathway (figure 2). Usual care pathways follow NICE guidelines [28] and patients will undergo routine, usually annual, pacemaker follow-up. Interventional pathways consists of enhanced or optimised care pre-determined by each recruiting site. The echocardiographically informed pathways will consist either of a primary care driven management plan based upon the echocardiogram which will be forwarded to the primary care team (enhanced care, n=300) if recruited in one of the district centres, or a comprehensive care package consisting of optimised programming and medical therapy co-ordinated by a regional heart failure service (optimised care, n=300) if recruited in the tertiary centre. This interventional package will include the RV pacing avoidance algorithm (figure 1) and in those with LVSD (LVEF<50%) medical therapy following the NICE guidelines for the management of chronic heart failure [29]. In the presence of LVSD, patients recruited at the tertiary centre will be referred to an outpatient HF clinic (optimized care), and in the district centres an echocardiogram report will be sent to the patients’ general practitioner (enhanced care). “

Referee 2:

We are grateful for the supportive and in-depth review of the protocol and have made the following changes in response to your thoughtful comments which have improved the manuscript.

According to the information on ClinicalTrials.gov the trial started August 2013, the actual primary completion date was October 2017, and the latest recorded anticipated study completion date is this month (February 2019). The anticipated study completion date has been moved forward several times after the primary completion date. Publication of the protocol at this late stage when the trial is basically finished does not ensure increased transparency regarding deviation from the protocol which is a major reason for publishing a protocol.

The expert reviewer makes a good point. The completion date extensions are in part due to constraints of the fellowship funding start period, and due to the trial listed on clinicaltrials.gov also including an adjacent observational cohort study. We are committed to providing transparent detailed information with regards to the RCT protocol and have no trial deviations to report.

In the protocol the enrolment number is stated to be 1200 participants. According to information on ClinicalTrials.gov, 1793 patients were enrolled before completion (October 2017), but in earlier study record versions 2100 participants were anticipated. Is this discrepancy an error or was the sample size adjusted during the trial period?

Thank you for this comment. The total number of participants for the randomised trial sample is 1200. The additional participants and reason for the discrepancy in figures is due to the additional observational cohort included within the study information on Clinicaltrials.gov. We have reviewed the submitted protocol and ensured only the details of the randomised trial are included.

There is lack of detail in the protocol

We appreciate the reviewers comments and have clarified the points below and within the protocol text as highlighted.

It is not clear how it was decided which patients received enhanced care or optimised care in the intervention group.

Patients are stratified to interventional groups according to recruitment site. Patients at the two district centres randomised to intervention will receive enhanced care (n=300), and those at recruited at the tertiary centre and randomised to intervention are assigned to optimised care (n=300). We have clarified this in lines 217 to 223.

“Interventional pathways consists of enhanced or optimised care pre-determined by each recruiting site. The echocardiographically informed pathways will consist either of a primary care driven management plan based upon the echocardiogram which will be forwarded to the primary care team (enhanced care, n=300) if recruited in one of the district centres, or a comprehensive care package consisting of optimised programming and medical therapy co-ordinated by a regional heart failure service (optimised care, n=300) if recruited in the tertiary centre.”

What was end of follow up - how will data from patients who had heart transplants or changed to other CIEDs be analysed?

All patients will undergo a digital follow-up at 12 months to assess for the primary endpoint. We will collect and report those patients who go on to have cardiac transplants or device upgrades during this period. We have expanded on the information provided in line 274.

“Digital follow-up will occur at 12 month post recruitment to record mortality and hospitalisation data and changes to medical therapy. Patient data regarding cardiac transplatation or device upgrade will be recorded and subsequently reported.”

Is the primary outcome a combined end-point (time to death and heart failure hospitalisation) or either heart failure hospitalisation or death?

The primary outcome of the study is a combined end-point of time to death or first heart failure hospitalisation. We have made this more clear following your advice in line 239.

“The primary outcome measure is a combined endpoint of time from randomisation to date of first event of all-cause mortality or heart failure hospitalisation as assessed by the Endpoint Review Committee.”